# Blood Group and Response to Bariatric Surgery in Morbidly Obese Patients: A Retrospective Study in Saudi Arabia

**DOI:** 10.3390/healthcare11010052

**Published:** 2022-12-24

**Authors:** Albaraa H. Kazim, Fahad Bamehriz, Hamad Saud S. Alsubaie, Abdullah Aldohayan, Hussam Alamri, Abdallah Y. Naser, Al-bandari Zamil Abdullah, Lamis Mustafa Jaamour, Ghaida AlBraithen, Ghada Bamogaddam, Omar Mostafa

**Affiliations:** 1Alnoor Specialist Hospital, Makkah 24241, Saudi Arabia; 2Bariatric and UGI Surgery Unit, King Saud University, Riyadh 12372, Saudi Arabia; 3Department of Surgery, College of Medicine, King Saud University, Riyadh 12372, Saudi Arabia; 4Department of Applied Pharmaceutical Sciences and Clinical Pharmacy, Faculty of Pharmacy, Isra University, Amman 11610, Jordan; 5Department of Surgery, Alhabib Medical Group Hospital, Riyadh 12214, Saudi Arabia; 6College of Medicine, Almaarefa University, Riyadh 11597, Saudi Arabia; 7College of Medicine, King Saud University, Riyadh 12372, Saudi Arabia; 8College of Medicine, Sulaiman Al Rajhi University, Al Bukayriyah 52726, Saudi Arabia

**Keywords:** Rh blood group, bariatric surgery, laparoscopic sleeve gastrectomy, obesity

## Abstract

Objectives: To explore the relationship between the blood group of patients and their response to bariatric surgery and to identify predictors of better outcomes. Methods: This was a retrospective cross-sectional analysis of patients who underwent laparoscopic sleeve gastrectomy for morbid obesity between 2014 and 2020 at King Saud University Medical City in Riyadh, Saudi Arabia. Results: This study included 1434 individuals. The mean change in BMI (pre- versus post-BMI) differed statistically significantly between blood groups (*p* ≤ 0.01). The greatest drop in body weight was seen in individuals with the AB-negative blood type (56.0 (21.4) kg), which corresponds to the greatest percentage of reduction from baseline (47.7% (14.8)). The mean BMI of the patients decreased by 34.7% (9.2) from a mean pre-operation BMI of 45.5 (8.4) kg/m^2^ to 29.7 (6.1) kg/m^2^ (*p* ≤ 0.001). After laparoscopic sleeve gastrectomy, male patients and those with the B-negative blood type are more likely to see a greater BMI reduction (pre-operation compared to post-operation) (*p* ≤ 0.05). Conclusions: For morbidly obese patients, laparoscopic sleeve gastrectomy demonstrated promising weight loss outcomes. Blood groups may be able to predict the success rate of bariatric surgery in morbidly obese patients.

## 1. Introduction

Obesity is characterized by an abnormal or excessive deposition of body fat which could have negative health consequences [1]. According to the World Health Organization (WHO)’s classification of adult obesity, a body mass index (BMI) of 25 to 29.9 kg/m^2^ is considered overweight while a BMI above 30 kg/m^2^ is classified as obese [2]. One of the most significant modern risks to public health is the increase in obesity rates around the globe. In Saudi Arabia, the estimated national weighted prevalence of obesity (BMI ≥ 30 kg/m^2^) in 2021 was 24.7% [3]. For adults who are morbidly obese, weight loss surgery has been shown to be a reliable and effective treatment [4]. In the past, laparoscopic sleeve gastrectomy (LSG) was referred to as a potential first-stage therapy for obese patients, but it is now frequently performed as a stand-alone bariatric procedure for high-risk and morbidly obese patients [4]. The ABO blood group system is a classification system for human blood based on the presence or absence of particular surface markers on red blood cells. The four major blood types are A, B, O, and AB [5]. The Rh blood group system consists of approximately 50 antigens, but the D antigen is the most relevant since it is the most immunogenic. In standard transfusion terminology, a patient’s “positive” or “negative” status is determined by its presence or absence [6].

Previous research has shown that blood groups are particularly associated with a wide range of diseases [7,8]. Blood groups and obesity have been the subject of a small number of studies with wide a range of findings [9,10]. A previous Indian study found a correlation between blood group B and the frequency of obesity and hypertension [11]. However, a different study from India found a correlation between childhood obesity and the O blood group [9]. In addition, Suadicani and colleagues found that males with the O blood group who had long-term work exposure to a variety of respirable airborne pollutants had a higher prevalence of obesity than men with other blood groups. Blood groups and high BMI were previously investigated in a Saudi population, and it was found that there is no significant association between overweightness and obesity and blood groups [1]. According to a book by Dr. Peter D’Adamo that was previously published, blood types mirror individuals’ internal chemistry [12]. This book also emphasized how blood types affect individuals’ susceptibility to disease, the foods they should eat, and the techniques to prevent a variety of health issues. One of the key outcomes that varies depending on blood type is weight reduction because of a number of factors, including variations in digestive function across individuals with various blood groups [12]. Additionally, it has been shown that blood type influences the selection of the appropriate diet and lifestyle for certain genotypes, which can alter the expression of particular genetic signals that cause illness and activate the expression of messages that promote health [12]. At the same time, the “Blood-Type” diet concept is not supported by other research, which indicates that following specific “Blood-Type” diets has positive effects on several cardiometabolic risk factors. However, these relationships were not dependent on an individual’s ABO genotype [13]. To the best of our knowledge, there is no previous study that explored the association between blood group and the response to bariatric surgery. Therefore, the aim of this study was to explore the relationship between the patients’ blood groups and their responses to bariatric surgery and to identify predictors of a better outcome.

## 2. Method

### 2.1. Study Design

This was a retrospective cross-sectional study that used the data of patients who underwent laparoscopic sleeve gastrectomy for morbid obesity at King Saud University Medical City in Riyadh, Saudi Arabia for the duration between 2014 and 2020.

### 2.2. Study Population

All patients from all age groups, whether males or females, who underwent laparoscopic sleeve gastrectomy for morbid obesity formed the study population.

### 2.3. Data Extraction

For patients who meet the inclusion criteria, the following data were extracted from the electronic medical system of the participating hospital: patients’ demographics (age, gender, weight, height, BMI as preoperative measures, and ABO–Rh blood group) and clinically relevant information (one year post-operative BMI).

### 2.4. Statistical Analysis

Data were analyzed using the SPSS software, version 27. Means (standard deviation [SD]) were used to report descriptive results. Kolmogorov–Smirnov test, Shapiro–Wilk test, and histograms were used to check the data’s normality. Categorical data were reported as percentages (frequencies). The BMI improvement was presented as a percentage change from the baseline measurement before the laparoscopic sleeve gastrectomy. Body weight before and after laparoscopic sleeve gastrectomy was compared by a paired *t*-test (dependent sample *t*-test). An ANOVA test was used to examine the significance of the difference in the mean change in BMI (pre- versus post-BMI) between different blood groups. Binary logistic regression was used to identify predictors of better BMI reduction. The dummy variable for the logistic regression was defined as a BMI change greater than 15.7%, which is the mean change of the study sample. The level of significance was assigned as 5%.

## 3. Results

### 3.1. Patients’ Baseline Characteristics

A total of 1434 patients who underwent laparoscopic sleeve gastrectomy were included in this study. More than half of them (54.8%) were females. The mean age of the patients was 34.6 (11.6) years. The mean pre-operation body weight was 123.3 (26.1) kg. The mean patients’ height was 164.5 (9.7) cm. The mean baseline BMI was 45.5 (8.4) kg/m^2^. Almost half of the patients (48.0%) were of the A-positive blood group. For further details on the baseline characteristics of the patients, refer to Table 1.

### 3.2. The Effect of Laparoscopic Sleeve Gastrectomy on Body Weight Stratified by Blood Group Type

The mean body weight of the patients who underwent laparoscopic sleeve gastrectomy decreased by 42.8 (16.9) kg from a mean pre-operation weight of 123.3 (26.1) to 80.5 (17.7) kg post-operation, (*p* ≤ 0.001). At baseline, there was no statistically significant difference in pre-BMI among the patients of different blood group types (*p* ≥ 0.05). There was a statistically significant difference in the mean change in BMI (pre- versus post-BMI) among different blood groups (*p* ≤ 0.01). Patients with the AB-negative blood type had the greatest reduction in body weight (56.0 (21.4) kg), representing the greatest percentage reduction from baseline (47.7% (14.8) reduction from baseline). The mean BMI of the patients decreased by 34.7% (9.2) from a mean pre-operation BMI of 45.5 (8.4) kg/m^2^ to 29.7 (6.1) kg/m^2^ (*p* ≤ 0.001). The highest reduction in BMI was observed in AB-negative patients followed by B-negative patients, O-positive patients, O-negative patients, A-positive patients, B-positive patients, AB-positive patients, and then A-negative patients. For further details on changes in body weight and body mass index stratified by blood group type, refer to Table 2.

Table 3 below presents the percentage change in BMI stratified by gender and blood group type. Overall, males showed a higher percentage reduction in BMI after the operation compared to females. Among male patients, the percentage reduction in BMI ranged between 32.9% and 66.1%, and those with the AB-negative blood type showed a higher percentage reduction. Among female patients, the percentage reduction in BMI ranged between 31.5% and 38.7%; those with the B-negative blood type showed a higher percentage reduction.

### 3.3. Predictors of Higher BMI Reduction

Male patients and those with the B-negative blood type are more likely to have a higher BMI reduction (pre-operation vs. post-operation) after laparoscopic sleeve gastrectomy according to binary logistic regression analysis (*p* ≤ 0.05), Table 4.

## 4. Discussion

A previous large-scale systematic review reported that, in comparison to non-surgical treatment, bariatric surgery was more cost-effective for moderately to severely obese patients [14]. Today, bariatric surgery is the most effective treatment for weight loss and comorbidity burden reduction in individuals with severe obesity. However, not all patients benefit equally from bariatric surgery. Predicting reactions to bariatric surgery could be a valuable tool in clinical practice either by enhancing patient selection or by identifying patients who require more active follow-up and postoperative management [15]. Our study aimed to explore the relationship between the patients’ blood groups and their responses to bariatric surgery and to identify predictors of a better outcome. The key findings are as follows: (1) the mean decrease in body weight of the patients who underwent laparoscopic sleeve gastrectomy was 35.0%; (2) the reduction in body weight was different across patients of different blood group types; and (3) males and patients with B-negative blood types are more likely to obtain a higher BMI reduction after undergoing laparoscopic sleeve gastrectomy.

In our study, the mean body weight of the patients who underwent laparoscopic sleeve gastrectomy decreased by 42.8 (16.9) kg from a mean pre-operation weight of 123.3 (26.1) to 80.5 (17.7) kg post-operation (*p* ≤ 0.001). The mean BMI of the patients decreased by 34.7% (9.2) from a mean pre-operation BMI of 45.5 (8.4) kg/m^2^ to 29.7 (6.1) kg/m^2^ (*p* ≤ 0.001). Gloy et al. observed in a previous meta-analysis that bariatric surgery resulted in more weight loss than non-surgical treatment (mean difference of −26 kg (95% confidence interval of −31 to −21)) [16]. Additionally, bariatric surgery was associated with greater rates of type 2 diabetes and metabolic syndrome remission [16]. Laparoscopic sleeve gastrectomy is a restrictive treatment in which approximately 80% of the stomach is excised, resulting in additional hormonal regulation that helps patients alter their eating habits without major morphological or functional change [17]. It has also been demonstrated that LSG is less expensive than other bariatric operations and treatments of comorbidities and problems in non-operated patients [18,19].

Numerous prior research studies have demonstrated the significance of the blood type system in defining illness and health conditions [20,21,22]. The literature on the correlation between blood group and BMI are contradictory [23]. Recent research investigated the link between blood grouping and obesity [11,24]. In a previous study conducted in Brazil with 549,690 individuals, Flor et al. found a gender-specific link between particular ABO antigens and BMI [25]. The O and B blood types were connected with a higher frequency of obesity in females compared to males. Blood types B and O were associated with a higher frequency of obesity in females. The O and B blood types were associated with a lower incidence of obesity in males [25]. Another study by Chandra et al. reported that patients with blood group type B followed by blood type O patients were more likely to be obese compared to others [11]. The blood group system is regarded as one of the most accurate predictors of obesity-related genetic markers [24,26]. As previously indicated, the association between obesity and the blood group system has been studied; however, to our knowledge, no studies have examined the blood group system as a predictor of weight reduction after bariatric surgery. Blood type influences the choice of diet and lifestyle for certain genotypes, which can alter the expression of specific genetic signals that cause disease and stimulate the expression of genetic messages that promote health and contribute to variations in digestive function across individuals with various blood groups [12].

In our study, the largest reduction in body weight was observed among patients with the AB-negative blood type, and males and patients with the B-negative blood type are more likely to have a higher BMI reduction (pre-operation compared to post-operation) after undergoing laparoscopic sleeve gastrectomy. One possible justification for the gender-based difference is that male patients also demonstrated greater weight loss than female patients, which can be attributed to their greater muscle mass and activity levels [27,28]. As a result of physiological and psychological gender differences, it has also been observed that males lose more weight than females [28].

Our study has multiple strengths. Our study population included patients who underwent LSG only. Selecting patients who underwent sleeve gastrectomy alone has the advantage of reducing bias from selecting different procedures with different efficacies. Sleeve gastrectomy is the most common procedure performed today in the world, particularly in the Middle East, with comparable results to gastric bypass and lower morbidity than gastric bypass [29,30]. One of the main strengths of our study is the large sample size of the study sample. At the same time, this study has limitations. Only six patients who had blood group type AB-negative were involved in this study, which might not have a clear estimate of the weight reduction among this group of patients. There were no data available on body composition analyses for the study cases, which restricted our ability to determine which parts of the body lost more weight. There were no data on other co-existing diseases (comorbidities) or obesity indices, which might have restricted our ability to define other contributing factors that might have affected patients’ outcomes. However, the relevance of these findings in weight reduction calls for a larger study population or one that includes more patients with the AB-negative group to confirm this association.

Future research on a larger scale and representing other demographic groups are required to verify our findings and identify other factors associated with greater BMI loss after bariatric surgery in response to different blood types. In addition, additional research examining the effect of various weight loss strategies on post-intervention weight loss is required.

## 5. Conclusions

Laparoscopic sleeve gastrectomy showed promising findings related to weight reduction in morbidly obese patients. Males and patients with the B-negative blood type are more likely to have a higher BMI reduction after undergoing laparoscopic sleeve gastrectomy. Blood groups could be a potential predictor of bariatric surgery success rates in morbidly obese patients. Future studies are warranted to confirm our study’s findings.

## Figures and Tables

**Table 1 healthcare-11-00052-t001:** Patients’ baseline characteristics.

Variable	Frequency	Percentage
Gender		
Females	786	54.8%
Mean age (standard deviation) years	34.6 (11.6) years	
Weight (Kg)	123.3 (26.1) Kg	
Height (cm)	164.5 (9.7) cm	
BMI (kg/m^2^)	45.5 (8.4) kg/m^2^	
Blood group type		
O positive	688	48.0%
A positive	332	23.2%
B positive	243	16.9%
O negative	64	4.5%
AB positive	49	3.4%
A negative	27	1.9%
B negative	25	1.7%
AB negative	6	0.4%

**Table 2 healthcare-11-00052-t002:** Change in body weight and body mass index stratified by blood group type.

Variable	Pre-Operation Weight	Post-Operation Weight	Total Weight Loss	*p*-Value	Pre-Operation BMI	Post-Operation BMI	Total BMI Loss	*p*-Value	% Change in BMI	*p*-Value (between Different Blood Groups)
All patients	123.3 (26.1)	80.5 (17.7)	42.8 (16.9)	≤0.001 *	45.5 (8.4)	29.7 (6.1)	15.7 (5.7)	≤0.001 *	−34.7% (9.2)	
A-positive patients	121.8 (28.3)	80.1 (19.5)	41.7 (17.1)	≤0.001 *	45.0 (8.9)	29.7 (6.6)	15.4 (5.8)	≤0.001 *	−33.7% (9.5)	≤0.01
A-negative patients	124.7 (28.2)	84.7 (20.0)	40.0 (14.0)	≤0.001 *	45.3 (10.0)	30.6 (6.7)	14.8 (5.3)	≤0.001 *	−32.2% (7.3)	
B-positive patients	122.2 (26.5)	80.7 (18.6)	41.5 (18.0)	≤0.001 *	44.9 (8.2)	29.7 (6.4)	15.2 (5.9)	≤0.001 *	−33.5% (10.1)	
B-negative patients	123.7 (20.4)	75.7 (13.5)	48.0 (13.5)	≤0.001 *	45.3 (6.8)	27.6 (3.9)	17.7 (5.1)	≤0.001 *	−38.6% (7.0)	
AB-positive patients	124.1 (25.4)	81.9 (15.7)	42.2 (16.3)	≤0.001 *	45.6 (8.2)	30.1 (5.1)	15.5 (5.6)	≤0.001 *	−33.4% (8.5)	
AB-negative patients	131.0 (17.2)	74.9 (13.6)	56.0 (21.4)	≤0.001 *	46.8 (4.8)	24.9 (8.8)	19.7 (5.1)	≤0.001 *	−47.7% (14.8)	
O-positive patients	124.5 (25.1)	80.8 (16.6)	43.7 (16.7)	≤0.001 *	46.0 (8.2)	29.9 (5.9)	16.1 (5.7)	≤0.001 *	−34.6% (9.0)	
O-negative patients	119.7 (25.4)	78.4 (16.9)	41.3 (15.5)	≤0.001 *	44.7 (7.6)	29.4 (5.5)	15.3 (5.0)	≤0.001 *	−33.9% (8.6)	

* *p* ≤ 0.001.

**Table 3 healthcare-11-00052-t003:** Percentage change in BMI stratified by gender and blood group type.

Variable	% Change in BMI	
	Males	Females
All patients	−36.2% (9.8)	−32.5% (8.5)
A-positive patients	−35.9% (10.1)	−32.1% (8.8)
A-negative patients	−32.9% (8.1)	−31.5% (6.8)
B-positive patients	−35.5% (10.4)	−31.7% (9.4)
B-negative patients	−38.6% (7.6)	−38.7% (6.4)
AB-positive patients	−34.0% (8.4)	−32.6% (8.7)
AB-negative patients	−66.1% (4.1)	−38.5% (4.5)
O-positive patients	−36.5% (9.5)	−32.9% (8.1)
O-negative patients	−36.6% (8.8)	−32.1% (8.0)

**Table 4 healthcare-11-00052-t004:** Binary logistic regression analysis.

Variable	Odds Ratio of Higher BMI Reduction	*p*-Value
Gender		
Females (reference group)	1.00	
Males	2.2 (1.8–2.7)	≤0.001 **
Mean age (standard deviation) years	0.97 (0.96–0.98) **	
Blood group type		
A positive (Yes)	1.2 (0.9–1.5)	0.175
A negative (Yes)	0.8 (0.4–1.7)	0.579
B positive (Yes)	0.9 (0.7–1.2)	0.588
B negative (Yes)	3.1 (1.3–7.4)	0.012 *
AB positive (Yes)	0.8 (0.5–1.4)	0.457
AB negative (Yes)	5.9 (0.7–50.6)	0.106
O positive (Yes)	1.1 (0.9–1.4)	0.322
O negative (Yes)	1.0 (0.6–1.7)	0.889

* *p* ≤ 0.05, ** *p* ≤ 0.001.

## Data Availability

Supporting data are available on request.

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
