# Peer review of "Blood Group and Response to Bariatric Surgery in Morbidly Obese Patients: A Retrospective Study in Saudi Arabia"

_healthcare, 2022, doi:10.3390/healthcare11010052_

Round 1
Reviewer 1 Report
Generally |
The authors studied the association of the ABO system with weight loss after BS in a huge sample size via retrospective analysis. However, the study is merely descriptive without any basis for the suggested association. The English language needs editing. |
Abstract |
· Based on author guidelines abstract should be limited to 200 words · The rationale should be clarified. Why authors selected ABO blood grouping as a predictor for weight loss after BS, and what is the clinical value? · Despite indicating the ABO system for blood grouping, the results mentioned the Rh system. Please revise |
Introduction |
· The authors should indicate if the ABO system or ABO-Rh system was used for blood grouping. · Authors should enumerate some studies that linked weight loss and blood grouping even in lifestyle-induced weight loss. |
Methodology |
· How do you measure Blood groups? · Are body composition analyses done for cases? It was valuable to know which compartment is losing more. · Were other anthropometric measures reported such as Waist circumference, percentage of excess weight loss, etc? Authors depend only on BMI which is proven as a surrogate indicator of obesity. |
Results |
· In table 1: What is the value of reporting H Pylori infection? · Changes in comorbidities among blood groups Deserve reporting
|
Discussion |
· 1st part of the discussion was focusing on the value of BS in the management of obesity, which is a well-known finding and not related to the aim of this work. · It is required to discuss the relationship between blood grouping and weight loss even in nonsurgical causes. · Possible mechanisms that support the main findings of this work should be suggested and discussed. · What is the clinical value of the study findings?
|
Bibliography/References |
Okay |
Final Note |
Major revision |
Author Response
Generally |
The authors studied the association of the ABO system with weight loss after BS in a huge sample size via retrospective analysis. However, the study is merely descriptive without any basis for the suggested association. The English language needs editing. - We have now checked the language of the manuscript. - First of all, we would like to thank the reviewer for the time and efforts in reviewing our manuscript. The aim of our study was to explore the relationship between the patients’ blood group and their response to bariatric surgery and to identify predictors of a better outcome. We have conducted logistic regression to identify predictors of better BMI reduction. |
Abstract |
· Based on author guidelines abstract should be limited to 200 words - Thank you for this comment, we have now reduced the words count in the abstract to address the reviewer comment. · The rationale should be clarified. Why authors selected ABO blood grouping as a predictor for weight loss after BS, and what is the clinical value? - Thank you for this comment, the blood grouping and its relations to different Human diseases and the risk of future diseases is studied in the literature and ongoing further studies is going on it, and as we are observing different response to bariatric surgeries among the patient underwent the same procedure we started to think about the factors that affect the weight loss and the response to BS. In this paper we studied and analyse this factor and we recommend further research in this factor with larger or with national database if possible. The clinical value of this type of research is that blood groups may be able to predict the success rate of bariatric surgery in morbidly obese patients · Despite indicating the ABO system for blood grouping, the results mentioned the Rh system. Please revise - Thank you for this comment, we have now addressed the reviewer comment in the method section (data extraction section). |
Introduction |
· The authors should indicate if the ABO system or ABO-Rh system was used for blood grouping. Thank you for this comment, the ABO-Rh system was used and is now highlighted in the method section. · Authors should enumerate some studies that linked weight loss and blood grouping even in lifestyle-induced weight loss. Thank you for this comment, the literature is full of Blood-Type special diet and its hypothesis and such as (Eat Right 4 your type ) and the published articles in trying prove and disprove the effectiveness of this special blood type diet and the metabolic result. However, to the best of our knowledge there no studies linked non surgical weight loss and Blood group except the studies of blood type diet. We have now highlighted these points further in the introduction section.
|
Methodology |
· How do you measure Blood groups? Rh typing and ABO blood grouping procedures: Making a 10% RBC suspension in regular saline- 5 drops (each 50 µl) of sedimented RBCs should be combined with 2 ml of ordinary saline. Spend one minute centrifuging at 1500 RPM, then remove the supernatant. Then, thoroughly combine 2 ml of ordinary saline with the sedimented RBCs. The RBC suspension produced by this preparation is 10%. Place a drop of antisera-A on one half of a glass slide and a drop of antisera-B on the other. A drop of antisera-D should be placed in the middle of a different glass slide. Add a drop of the RBCs suspension to each individual antiserum using a pipette or dropper. Making use of different applicator sticks, combine each cell with antisera. Slide the glass back and forth for as long as two minutes. Keep an eye out for clumping or agglutination. Results Interpretation for Rh Typing and ABO Blood Grouping pertaining to the ABO blood group system:
Antisera-A clumping but not antisera-B: blood group A
Blood type B exhibits clumping or agglutination on antisera B but not A.
Antisera A and B clumping: Blood type AB
Blood group O: No clumping on both antisera-A and B
For typing Rh
Positive clumping on antisera-D
Antisera-D did not clump: negative Are body composition analyses done for cases? It was valuable to know which compartment is losing more. It’s interesting to know further details but as the study is retrospective the data collected from center has no such facilities, we have now added these details to the study limitations · Were other anthropometric measures reported such as Waist circumference, percentage of excess weight loss, etc? Authors depend only on BMI which is proven as a surrogate indicator of obesity. -Thank you for this comment, we totally agree with the reviewer concerning this point that waist and neck circumference is indicator for obesity and risk of obesity related disease, however, in this medical center they are depending on patient’s weight and BMI as indication for BS. |
Results |
· In table 1: What is the value of reporting H Pylori infection? · Changes in comorbidities among blood groups Deserve reporting -Thank you for this comment, based on previous literature, H. Pylori infection was proposed to be associated with weight gain, therefore, we were interested to investigate whether it is affecting the BMI reduction after bariatric surgery among our study population. This study was a retrospective study that obtained patients data from their medical records. Unfortunately, we did not have information on changes in comorbidities among blood groups. |
Discussion |
· 1st part of the discussion was focusing on the value of BS in the management of obesity, which is a well-known finding and not related to the aim of this work. · Thank you for this comment, in the first paragraph of the discussion we aimed to give the reader a background information about the relationship between blood grouping and weight loss even in nonsurgical causes to engage the reader in our research topic and let him/her understand and interpret our findings more preciously. · Possible mechanisms that support the main findings of this work should be suggested and discussed. - Thank you for this comment, we have now added Possible mechanisms that support the main findings of this work in the discussion section in page 5. · What is the clinical value of the study findings? Clarification of factors affecting the weight loss following BS is important for treatment journey for obese patient affecting the decision and expectation of the surgery, weighing the risk and the benefit of the surgery on each individual going for it. The clinical value of this type of research is that blood groups may be able to predict the success rate of bariatric surgery in morbidly obese patients. We have now highlighted these points in the introduction section. |
Reviewer 2 Report
Thank the editors for the opportunity to review. This is a very interesting, innovative analysis of the results of bariatric surgery. It should be treated more as a scientific curiosity than a guideline, but I think it should be published. However, a few minor things are missing.
Were the patients, apart from the blood group, significantly different in something that could also affect weight loss? Comorbidities?
AB negative and 6 patients are not enough to consider this blood group as likely in a good prediction of weight loss. It was rightly noticed in the limits, but it should not be a summary of the results (one of the 3 points in the first paragraph of the discussion)
The conclusions lack the conclusions from the work that I found in the discussion.
Author Response
Thank the editors for the opportunity to review. This is a very interesting, innovative analysis of the results of bariatric surgery. It should be treated more as a scientific curiosity than a guideline, but I think it should be published. However, a few minor things are missing.
Were the patients, apart from the blood group, significantly different in something that could also affect weight loss? Comorbidities?
- First of all, we would like to thank the reviewer for the time and efforts in reviewing our manuscript. Thank you for this comment, actually, according to the available data at the medical records of the patients, the patients were very similar to each other in term of their baseline characteristics, all of them were young with a mean age of 34.6 years and almost half of them were males.
AB negative and 6 patients are not enough to consider this blood group as likely in a good prediction of weight loss. It was rightly noticed in the limits, but it should not be a summary of the results (one of the 3 points in the first paragraph of the discussion).
- Thank you for this comment, based on the reviewer comment, we have now changed the second point in the key findings mentioned in the first paragraph of the discussion to be “reduction in body weight was different across patients from different blood group types”.
The conclusions lack the conclusions from the work that I found in the discussion.
- Thank you for this comment, we have now highlighted further the main conclusion from our work in the conclusion section.
Reviewer 3 Report
Manuscript ID healthcare-2061512
Title: Blood group and response to bariatric surgery in morbidly obese patients: a retrospective study in Saudi Arabia
Dear Authors,
I have reviewed your article,I am suggesting the following things for the betterment of your article.
1. Spelling and punctuation errors must be taken care.
2. Authors can improve the results presentation.
3. This article needs language correction.
4. Kindly, highlight which blood group had greatest susceptibility to be obese at appropriate places in the manuscript.
5. Mention the order of BMI reduction in the studied blood groups.
6. Authors can highest the highest and lowest prevalence of obesity in terms of ABO association in the conclusion since the title conveys about blood group and response to bariatric surgery in morbidly obese patients.
7. For men, which blood type (or types) was associated with a higher susceptibility, likewise for women? kindly describe this clearly in the results.
8. Authors have given details about obesity in the introduction, likewise could add a small introductory note about the ABO blood group rather than going straight away to the earlier researches on that.
Author Response
in Saudi Arabia
Dear Authors,
I have reviewed your article,I am suggesting the following things for the betterment of your article.
First of all, we would like to thank the reviewer for the time and effort in reviewing our manuscript.
Spelling and punctuation errors must be taken care.
- Thank you for this comment, we have now addressed this comment across the manuscript.
Authors can improve the results presentation.
- Thank you for this comment, we have now rephrased some paragraphs in the results section to make the flow better.
- This article needs language correction.
- Thank you for this comment, we have now addressed this comment across the manuscript.
Kindly, highlight which blood group had greatest susceptibility to be obese at appropriate places in the manuscript.
- Thank you for this comment, after comparing the mean BMI before the surgery, we found that there was no statistically significant difference between the patients, which did not support that specific blood group had greatest susceptibility to be obese. We have now added these information to the results section (section 3.2).
5. Mention the order of BMI reduction in the studied blood groups.
- Thank you for this comment, we have now added these information to the results section (section 3.2).
6. Authors can highest the highest and lowest prevalence of obesity in terms of ABO association in the conclusion since the title conveys about blood group and response to bariatric surgery in morbidly obese patients.
- Thank you for this comment, we have now highlighted further in the conclusion section the patients who are more likely to get benefit from laparoscopic sleeve gastrectomy in relation to their blood group type. In addition, based on the reviewer comment, we have now highlighted in the results section that we found that there was no statistically significant difference between the patients in their baseline BMI, which did not support that specific blood group had greatest susceptibility to be obese.
7. For men, which blood type (or types) was associated with a higher susceptibility, likewise for women? kindly describe this clearly in the results.
- Thank you for this comment, we have now addressed the reviewer comment and add Table 3 to highlight percentage change in BMI stratified by gender and blood group type in the results section.
8. Authors have given details about obesity in the introduction, likewise could add a small introductory note about the ABO blood group rather than going straight away to the earlier researches on that.
- Thank you for this comment, we have now addressed the reviewer comment and added a small introductory paragraph about the ABO blood group in the introduction section.
Round 2
Reviewer 1 Report
many comments could not be fixed in the revised form due to the perspective nature of this work and the data loss. however, it still points to a novel finding and could be accepted. all of these comments should be stressed in study limitations especially the lack of analysis of comorbidities, body composition changes, and other obesity indices, ..etc.
besides, the addition of H.Pylori to this work is not logical and the authors did not justify the relationship with the study objective. why is only H. pylori with no mention of other commodities used? Accordingly, it should be deleted.
Author Response
many comments could not be fixed in the revised form due to the perspective nature of this work and the data loss. however, it still points to a novel finding and could be accepted. all of these comments should be stressed in study limitations especially the lack of analysis of comorbidities, body composition changes, and other obesity indices, ..etc.
- Thank you for this comment, we have now added the above-mentioned limitations based on the reviewer comment.
besides, the addition of H.Pylori to this work is not logical and the authors did not justify the relationship with the study objective. why is only H. pylori with no mention of other commodities used? Accordingly, it should be deleted.
- Thank you for this comment, we have now deleted all findings related to H.Pylori based on the reviewer comment.